# A Reference-free Segmentation Quality Index (SegReFree)

**Evan Lucas** and **Dylan Kangas** and **Timothy C Havens**
Michigan Technological University / 1400 Townsend Drive
Houghton, Michigan, United States of America
{eglucas, dylank, thavens} @mtu.edu

## Abstract

Topic segmentation, in the context of natural language processing, is the process of finding boundaries in a sequence of sentences that separate groups of adjacent sentences at shifts in semantic meaning. Currently, assessing the quality of a segmentation is done by comparing segmentation boundaries selected by a human or algorithm to those selected by a known good reference. This means that it is not possible to quantify the quality of a segmentation without a human annotator, which can be costly and time consuming. This work seeks to improve assessment of segmentation by proposing a reference-free segmentation quality index (SegReFree). The metric takes advantage of the fact that segmentation at a sentence level generally seeks to identify segment boundaries at semantic boundaries within the text. The proposed metric uses a modified cluster validity metric with semantic embeddings of the sentences to determine the quality of the segmentation. Multiple segmentation data sets are used to compare our proposed metric with existing reference-based segmentation metrics by progressively degrading the reference segmentation while computing all possible metrics; through this process, a strong correlation with existing segmentation metrics is shown. A Python library implementing the metric is released under the GNU General Public License and the repository is available at https://github.com/evan-person/reference_free_segmentation_metric.

## 1 Introduction

Text segmentation is a common task in language processing, in which a text is split into segments. Text segmentation is performed at different scales: when it is used to split sentences into their parts, it is referred to as *elementary discourse unit* (EDU) separation, and when used to split larger documents into topic based groups of sentences, it is known as topic segmentation (Marcu and Echihabi, 2002; Beeferman et al., 1999). This work focuses on topic segmentation, which has many applications from information retrieval (Dias et al., 2007) to summarization of long documents (Gidiotis and Tsoumakas, 2020; Zhang et al., 2022).

Segmentation is generally evaluated by comparing a reference set of possible segment boundaries with the boundaries chosen by a given segmentation algorithm. This means that segmentation evaluation can only be performed on a data set that has already been segmented by a human evaluator (or other accepted-as-truth segmentation). A variety of methods have been proposed for evaluating segmentation, all of which focus on the reference and candidate sets, rather than the content being segmented. A review of these methods is included in Section 2. We significantly push the state-of-the-art in segmentation evaluation by proposing a reference-free segmentation quality metric.

The rest of this paper is organized in the following manner. A brief history of modern segmentation metrics and other related work is described in Section 2. The proposed reference-free segmentation quality metric is developed, alongside design choice explanations, in Section 3. Validation of the method, by comparing results to existing segmentation metrics on popular segmentation data sets are described in Section 4 along with an ablation study to demonstrate the behavior of the method in several scenarios. Potential uses and impacts of this work are presented alongside a summary in Section 5.

## 2 Related Work

Existing segmentation evaluation metrics, proposed by Beeferman et al. (1999); Pevzner and Hearst (2002); Fournier and Inkpen (2012); Fournier (2013) are reviewed, along with a brief introduction to cluster validity metrics. We include a review of the Davies-Bouldin Index (Davies and Bouldin, 1979), which was used as a starting point for this work. A brief review of some unsupervised

segmentation methods that use Sentence-BERT (Reimers and Gurevych, 2019) is also presented.

## 2.1 Existing Segmentation Metrics

A variety of segmentation evaluation metrics have been proposed (Beeferman et al., 1999; Pevzner and Hearst, 2002; Fournier and Inkpen, 2012; Fournier, 2013). To our knowledge, all currently used metrics focus on comparing the set of boundaries generated by the segmentation algorithm to some other set of boundaries, either a true reference or another hypothesis boundary set. Additionally, we are not aware of any existing segmentation quality evaluations that are based on cluster validity metrics or the semantic representations of the segmented text.

Existing segmentation metrics can be split into two categories, window-based and mass-based, with $P_k$ and WindowDiff being window-based and Segmentation Similarity and Boundary Similarity being mass-based. A simple visualization of these two ways of considering segmentation boundaries is presented in Appendix C. The SegEval (Fournier, 2013) implementation was used for the four classic segmentation metrics used in this work.

### 2.1.1 $P_k$

The $P_k$ metric was first proposed by Beeferman et al. (1999) and is widely used for evaluating segmentation quality (Arnold et al., 2019; Xia et al., 2022). $P_k$ is the conditional probability of a boundary classification error occurring within a window of $k$ chunks (making this a window based metric), given the reference and hypothesis boundary sets. Errors can be either a missing boundary or an added boundary within a moving window that goes across the segmentation set. As an error rate, a score of zero is the desirable score. Using Figure 8 as a reference, with the example window in the position shown, $P_k$ would not consider there to be an error present. The window is then transposed across the entire segmentation set, with errors calculated at each transposition position, and a combined score is created to represent the full segmentation.

### 2.1.2 WindowDiff

The $P_k$ metric, although widely used, has some noted shortcomings that were raised by Pevzner and Hearst (2002) in the development of their proposed metric, WindowDiff (Pevzner and Hearst, 2002). They observed that $P_k$ penalizes false negatives more highly than false positives, tends to

penalize transposed boundaries (known as near misses) too heavily, and is impacted by segment size variation. WindowDiff is simple to implement and compares the number of boundaries within a moving window. This metric is also widely used in the segmentation literature (Mota et al., 2019; Zhong et al., 2022). Similar to $P_k$, a score of zero is seen as desirable. Again, using Figure 8 as a reference, with the window in the example position shown, the number of boundaries present match and this window position would return a score of zero.

### 2.1.3 Segmentation Similarity

Fournier and Inkpen (2012) noted that WindowDiff and $P_k$ are both window-based metrics that depend on a reference segmentation, which introduces some issues. They noted that neither metric penalizes error types (missing or additional boundaries) equally and that window size influences the metrics' outcome greatly. They proposed a new metric called *segmentation similarity*, which attempted to improve upon existing metrics by approximating the distance between segmentation sets in terms of the number of edits required to make them the same. This method has the advantage of being symmetric (neither of the compared segmentation sets are treated as the true reference) and can be extended to compare segmentations between multiple annotators. Unlike $P_k$ and WindowDiff, segmentation similarity is a similarity measurement and a score of 1 is considered to be a perfect match.

### 2.1.4 Boundary Similarity

Fournier (2013) observed that their proposed segmentation similarity metric suffered from optimistic values and proposed a new set of metrics derived from the same motivations as segmentation similarity in Fournier (2013). The primary metric proposed is called boundary similarity, although a boundary edit distance based confusion matrix and metrics based on that confusion matrix are also proposed in this work. Similar to segmentation similarity, a score of 1 is considered optimal for boundary similarity.

## 2.2 Alignment based similarity

Diaz and Ouyang (2022) propose a new method of segmentation metrics based on the alignment of segments between a reference and hypothesis segmentation set. This method was not used as a comparison method for this paper.

## 2.3 Cluster validity metrics

Cluster validity metrics are a more generalized approach for assessing quality of segmentation in that they were designed to assess the quality of partitions found by unsupervised learning algorithms. A variety of cluster validity metrics exist, but in order to meet the design goal of not requiring a reference, only cluster validity metrics that are reference-free (known in clustering literature as internal evaluation schemes) were considered. Three such metrics considered were the Davies-Bouldin Index (Davies and Bouldin, 1979), the Calinski-Harabasz Index (Caliński and Harabasz, 1974), and the Silhouette Score (Rousseeuw, 1987). All three of these metrics attempt to compare the relative distance (or dissimilarity) of the members of a cluster to the distance (or dissimilarity) between clusters, with varying differences in how they are computed. Silhouette Score was rejected due to a lack of sensitivity when overlapping clusters are present, which is a common situation when considering groups of textual semantic representations. Adaptations of the Calinski-Harabasz Index and Davies-Bouldin Index were both used in initial trials, but the adaptation of the Davies-Bouldin Index was found to better demonstrate differences in segmentation performance.

### 2.3.1 Davies-Bouldin Index

The Davies-Bouldin Index (Davies and Bouldin, 1979) is computed on a set of labeled points in some n-dimensional space in the following way. For each label $i$, a centroid of the members of that label is computed and the average distance between the segment centroid and the members of the segment are computed and stored as $S_i$. Any distance measure can be used, but Euclidian distance is seen frequently in the literature.

The distances between all pairs of centroids are computed and stored as $M_{ij}$. A ratio of pairwise intra-cluster distances and centroid distances is then computed as the following.

$$R_{ij} = \frac{S_i + S_j}{M_{ij}} \qquad (1)$$

The maximum value of $R_{ij}$ for each label $i$ is taken,

$$\hat{R}_i = \max\{R_{ij}\},$$

and the average of these over all segments is reported as the Davies-Bouldin index value,

$$DB = \frac{1}{N} \sum_{i=1}^{N} \hat{R}_i,$$

where $N$ is the number of labels. The maximum term in $\hat{R}_i$ means that the most similar cluster (in other words, the worse case clustering) to label $i$ is included in the final Davies-Bouldin index. The final score can be thought of as the ratio of intra-cluster distance to inter-cluster distance, averaged across the worse case pair of clusters for each label. A low score indicates that clusters are relatively compact and well-separated, whereas a high score indicates that clusters are large and/or overlap.

## 2.4 Unsupervised segmentation methods

Although not a metric, unsupervised segmentation methods are strongly related to the SegReFree metric and could be used in some similar ways. For example, an unsupervised segmentation method could be used to create a reference segmentation set, which could be then used with any of the segmentation metrics previously discussed. It should be noted that this would be a comparison between two different segmentation methods and still subject to the shortcomings of whatever segmentation metric was utilized.

Two specific segmentation methods that are somewhat similar to our proposed method are published by Solbiati et al. (2021) and Ghinassi (2021). Solbiati et al. (2021) uses Sentence-BERT (Reimers and Gurevych, 2019) to generate embeddings as an input for a modified TextTiling algorithm that attempts to detect topic changes based on variations in segment similarity. Ghinassi (2021) uses a variety of different sentence embedding models along with a modified version of TextTiling that is also based on segment similarity. In both cases, they rely on Sentence-BERT embeddings to create a semantic representation of the sentences to be segmented and attempt to use that semantic information to derive similarity based boundaries.

## 3 Method

Conceptually, our proposed metric seeks to quantify segmentation quality by assuming that the desired segmentation for a given text occurs at topical boundaries. If the segmentation boundary separates two different semantically different groups of text, the semantic vector representations of the sentences in those segments should generally be close to the other sentences within the segment and separated from the sentences in adjacent text segments.

At a high level, the proposed metric works by splitting text into sentences or utterances (the term

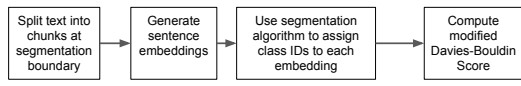

Figure 1: Metric computation flowchart

*chunks* is used as a synonym for both), generating embeddings of the chunks being segmented, and using a modified cluster validity metric where we treat segments as clusters. A block diagram is shown in Figure 1. Explanations of each step are elaborated upon in the rest of this section.

### 3.1 Text splitting

For the purposes of this paper and without loss of generality, all text was split at the sentence level following the previous usage of the data sets used for validation. This work could be extended to dialogue segmentation evaluation by splitting at the utterance level or it could be used for multi-document segmentation by splitting at a paragraph level.

### 3.2 Sentence embeddings

Sentence embeddings were generated using Sentence-BERT using the pre-trained 'all-mpnet-base-v2' model, which was trained on over one billion pairs of sentences from a variety of sources (Reimers and Gurevych, 2019). Sentence-BERT is a modification of the pretrained BERT network, which derives sentence embedding from semantic meanings and seeks to improve the vector space embedding over other methods of generating sentence level BERT embeddings. The goal of Sentence-BERT is to improve semantic textual similarity comparisons between sentences, which provides a necessary input for our proposed method. Sentence-BERT is passed sentences, from the text splitting, and returns 768 dimension embeddings, which are used in conjunction with labels as the input to the modified Davies-Bouldin Index discussed in 3.3. Other sentence embedding models could be used with the proposed method, as the SegReFree quality index can be used with any embedding that provides semantic representation of the content being segmented.

### 3.3 Modifications to the Davies-Bouldin Index

Two main modifications were required for the Davies-Bouldin Index to be used for assessment of segmentation quality. The first was restricting centroid distance calculations to adjacent segments.

This helps preserve the temporal aspect of segmentation, as assessing the quality of a boundary is best done by analyzing the adjacent segments to that boundary. Initial experiments found that the adjacent segment restriction improved sensitivity of the final SegReFree score. This effect can be conceptually explained quite easily: the quality of a boundary only depends on the segments being separated; a topically similar segment that isn't adjacent should not influence a descriptive score of segmentation quality. The impact of this change is most apparent when considering the error mode of missing boundaries, where adjacent segments get combined, which is presented in Section 4.1.1 and the impact of this modification is presented in the ablation study in Appendix D.

The second modification required was adding an exponential penalty term to the intra-segment distance computation. This was necessary due to the likelihood of adjacent chunks having a higher semantic similarity than non-adjacent chunks within a segment. Additionally, a segment that has a single sentence will have an intra-cluster distance of zero (ie. the centroid is the single point). This was found to be problematic during initial investigations where we created artificially low (good) scores by segmenting very frequently. The exponential penalty term was empirically created as a way to counteract this effect. We present this error mode in Section 4.1.2 and show the impact of this modification on smaller-than-desired segments in the ablation study in D. It is still possible to get a score of zero by treating each sentence as an independent segment, however this is an edge case and not an expected segmentation result. Our implementation of SegReFree will allow a user to score a segmentation with single sentence segments, but will warn them that it may cause an artificially low score. Multiple distance measures were considered (Euclidian, L1, and Cosine), with Euclidian being selected. Data supporting our choice of Euclidian distance is presented in the ablation study in Appendix D.

The SegReFree Index algorithm works as follows. Embeddings of all sentences and corresponding segment labels for all sentences are used as the input. For each segment, a centroid of the embeddings of the members of that segment is computed and the average Euclidian distance between the segment centroid and the members of the segment are computed; this average intra-cluster distance for

segment $i$ is referred to as dispersion $S_i$. These intra-cluster distances are then modified by

$$S_i = \frac{S_i}{1 - \frac{1}{\sqrt{n_i}}} \qquad (2)$$

for all segments containing more than one member, where $n_i$ is the number of members in the $i$th cluster. To prevent infinite values, the correction factor is set to 4 for segments where $n_i$ is equal to 1. The distances between centroids of temporally adjacent segments (e.g., between segment $i$ and both $i-1$ and $i+1$) are computed and stored as $M_{ij}, j \in \{i-1, i+1\}$. Non-temporally adjacent segment distances are not computed or compared, as they are not relevant to the segmentation boundaries. A ratio of pairwise intra-cluster distances and centroid distances is then computed as

$$R_{ij} = \frac{S_i + S_j}{M_{ij}}, \ j \in \{i-1, i+1\}, \qquad (3)$$

for both adjacent segments to $i$. The maximum value of $R_{ij}$ for each segment $i$ is taken,

$$\hat{R}_i = \max\{R_{i,i-1}, R_{i,i+1}\},$$

and the average of these over all segments is reported as the quality index value,

$$DB = \frac{1}{N} \sum_{i=1}^{N} \hat{R}_i,$$

where $N$ is the number of segments.

Essentially, $R_{ij}$ in (3) is the ratio of the added distances of segments $i$ and $j$ and $M_{ij}$, the distance between centroids of segments $i$ and $j$. If $R_{ij}$ has a low value then the segments are small compared to the distance between them. If $R_{ij}$ is large then the intra-cluster distances are large compared to the distance between them. Hence, the average of the maximum $R_{ij}$s for each segment $i$ represents a measure of the relative semantic dissimilarity of the the temporally adjacent segments are in a given segmentation.

### 3.4 Data sets

Two data sets commonly used in segmentation work were used to validate this metric. The first is the Choi data set, introduced by Choi (2000) and used to demonstrate the C99 segmentation algorithm. The Choi data set is constructed by taking paragraphs from different files in the Brown corpus

(Francis and Kucera, 1979), which creates substantial topical shifts between boundaries. The '3-11' subset of the Choi data set was used for this work, which includes paragraphs with three to eleven sentences each. Two files that included repeating paragraphs were excluded, as they show a very poor (high) score on the given segmentation boundaries between them and throw off average results. It should be noted that this is an artificially created data set that has strong topical shifts between segments and the results reported from this data set can be interpreted as close to ideal behavior of our proposed metric.

The second data set used is the newer Wiki-50 data set, which is scraped from Wikipedia and uses sections and subsections as topical segment boundaries (Koshorek et al., 2018). A larger Wiki-727k data set is also available from the same paper, however the Wiki-50 data set is sufficient to demonstrate the strengths and limitations of our proposed metric. Both data sets used in this work are freely shared for research purposes and the authors do not anticipate any issues with their inclusion in this work. Three other data sets derived from Wikipedia were also considered: the WikiSection (Arnold et al., 2019) and the Cities and Elements data sets (Chen et al., 2009). However, due to the similarity to the Wiki-50 data set they were not included in this work.

### 3.5 Computational infrastructure and budget

Sentence-BERT models were evaluated on either one or two A100 40GB GPUs and cluster metric computations were performed on CPU. The pre-trained 'all-mpnet-base-v2' Sentence-BERT model contains 109.5 million parameters. No models were trained for this work. Total computation time for this paper and related experimentation was less than 100 GPU hours.

## 4 Experiments

A variety of tests were devised to evaluate the performance of our metric. Tests were designed to mimic segmentation errors that could be encountered in realistic scenarios.

### 4.1 Metric performance on degraded segmentation sets

This series of trials all follow the same philosophy: to demonstrate how the metric reflects changes in an existing segmentation set as it is degraded

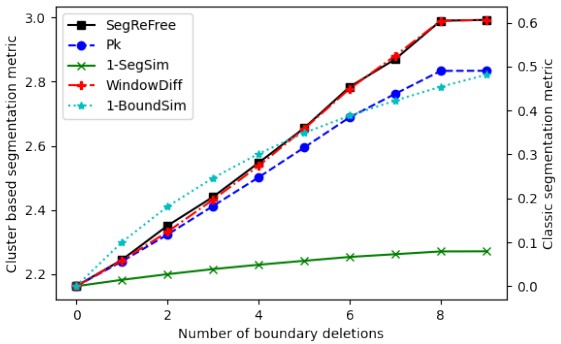

Figure 2: Proposed metric and classic metrics as a function of number of random boundary removals using the Choi 3-11 data set

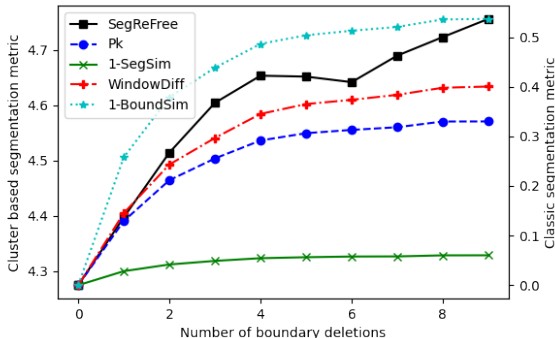

Figure 3: Proposed metric and classic metrics as a function of random boundary removals using the Wiki-50 data set

through different operations. For comparison, the four segmentation metrics described in Section 2 are included. All of these trials for each data set were performed in the same way, with the entire corpus being used and the results reported are the mean of all trials across the entire corpus. The Choi 3-11 corpus experiments were repeated five times and averaged to better sample the possible random outcomes and the Wiki-50 corpus experiments were repeated 30 times due to the smaller number of files. This brought the total number of averaged files for each corpus to be roughly the same (1500 for Wiki-50 and 1490 for Choi 3-11 due to the two excluded files).

### 4.1.1 Boundary removal

The first test performed was the removal of existing boundaries from a segmentation set. The existing boundaries were randomly selected and removed from each segmentation set. Classic reference-based metrics were computed using the original 'truth' and altered segmentation sets and the proposed metric was computed using only the altered segment labels (as ours is reference-free). Results for all 30 trials for the entire data set were averaged for each set of boundary deletions. If the metrics are effective, they should show that as boundaries are deleted and the segmentation changes from that of the 'truth' segmentation that the metric shows this degradation in segmentation quality.

Plots of mean proposed and classic metric values as a function of boundary removals are shown in Figures 2 and 3 for the Choi 3-11 and Wiki-50 data sets, respectively. To maintain vertical-axis consistency, the two similarity metrics are presented as one minus the metric; hence, higher values indicate

poorer segmentation quality. The strong correlation between existing metric values and SegReFree, as a function of increasingly degraded segmentation, can be clearly seen. The vertical-axis scales for the SegReFree index are set to make the most most efficient use of space and are therefore using different scales between Figures 2 and 3. The SegReFree index is not bounded to a consistent maximum value and should not be used to compare between models and datasets simultaneously. We suggest using it primarily for comparing segmentation methods on a consistent data set, although there may be applications where it is appropriate to use it to compare data sets. Conceptually, this can be explained by considering how different data sets use different parts of the embedding space.

Pearson correlation coefficients computed between classic reference-based metrics and our proposed reference-free metric are presented in Table 1. Note that the similarity metrics were converted to dissimilarity metrics for consistency: low score indicates good segmentation quality. It can be seen that there is a very strong agreement between the classic metrics and our reference-free metric, with correlation coefficients all 0.95. The trend is stronger with the Choi 3-11 data set than the Wiki-50, but this is expected due to the consistently larger topical shifts found in the Choi 3-11 data set.

### 4.1.2 Segment splitting

For our second experiment, we demonstrate the behavior of the segmentation quality indices when segments are randomly split, thus degrading the segmentation quality by adding boundaries. Segment splitting was performed by randomly select-

| Data Set | $P_k$ | Window-Diff | Seg. Sim.* | Bound. Sim.* |
|---|---|---|---|---|
| Choi 3-11 | 0.998 | 1.000 | 0.984 | 0.984 |
| Wiki-50 | 0.982 | 0.987 | 0.970 | 0.968 |

Table 1: Correlation Between Proposed Reference-Free Index and Reference-Based Indices in Boundary Removal Experiment

*Seg. Sim. and Bound. Sim. are converted to dissimilarity metrics for consistency.

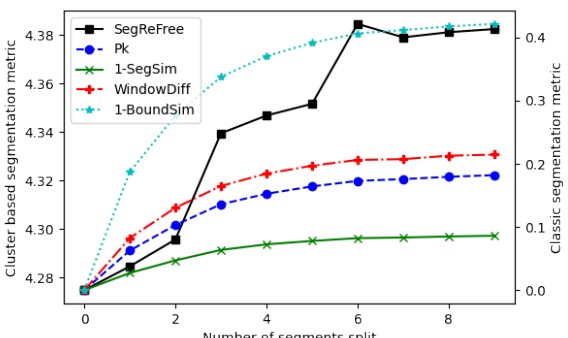

Figure 5: Proposed metric and classic metrics as a function of random segment splits using the Wiki-50 data set

| Data Set | $P_k$ | Window-Diff | Seg. Sim.* | Bound. Sim.* |
|---|---|---|---|---|
| Choi 3-11 | 0.998 | 0.999 | 0.989 | 1.000 |
| Wiki-50 | 0.939 | 0.924 | 0.946 | 0.902 |

Table 2: Correlation Between Proposed Reference-Free Index and Reference-Based Indices in Segment Splitting Experiment

*Seg. Sim. and Bound. Sim. are converted to dissimilarity metrics for consistency.

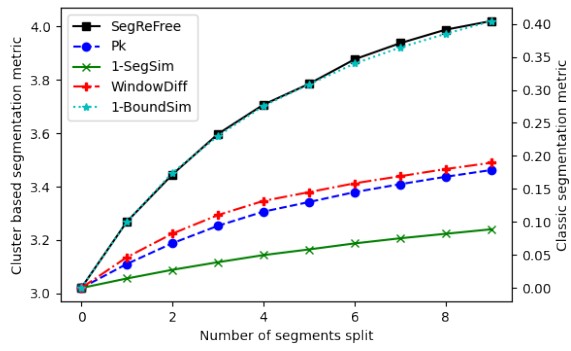

Figure 4: Proposed metric and classic metrics as a function of random segment splits using the Choi 3-11 data set

ing existing segments and splitting at the midpoint (rounding down in cases of odd numbers of sentences). As with the boundary removal experiment, the segmentation set was altered and the reference-based metrics were computed by comparing the altered segmentation with the original 'truth' segmentation. Only the altered segmentation was used to compute the proposed reference-free metric. The results of each were averaged and shown in Figures 4 and 5 for the Choi 3-11 and Wiki-50 data sets, respectively. Again, the two similarity metrics are presented as one minus the metric to maintain vertical consistency; where higher scores for all metrics indicate a lower quality segmentation. As with the boundary removal experiment, a strong correlation between existing metrics and SegReFree, as a function of increasingly degraded segmentation, can be clearly seen.

A table of Pearson correlation coefficients computed between the proposed metric and each of the classic metrics is presented in Table 2. A very strong correlation is observed for the Choi 3-11 data set: all $> 0.99$. The correlation between our proposed metric and the reference-based metrics

with the Wiki-50 data set is also good, although it is more inconsistent and almost completely insensitive to a single segment split. Thus, the overall correlation is not as strong in the Wiki-50 segment splitting experiment.

### 4.1.3 Boundary transposition

To simulate the condition of a 'near miss', as described in the segmentation metric literature, boundary transposition was also performed. In this experiment, an existing boundary was randomly se-

| Data Set | $P_k$ | Window-Diff | Seg. Sim.* | Bound. Sim.* |
|---|---|---|---|---|
| Choi 3-11 | 0.906 | 0.904 | 0.785 | 0.800 |
| Wiki-50 | -0.320 | -0.316 | 0.423 | 0.361 |

Table 3: Correlation Between Proposed Reference-Free Index and Reference-Based Indices in Boundary Transposition Experiment

*Seg. Sim. and Bound. Sim. are converted to dissimilarity metrics for consistency.

| Data Set | SegReF 'truth' | SegReF-TT | $P_k$ | Win. Diff | Seg. Sim. | B. Sim. | % correct |
|---|---|---|---|---|---|---|---|
| Choi 3-11 | 3.02 | 4.72 | 0.582 | 0.719 | 0.856 | 0.071 | (98%) |
| Wiki-50 | 4.27 | 4.84 | 0.485 | 0.494 | 0.819 | 0.161 | (68%) |

Table 4: Comparison of Segmentation Quality Metrics for Segmentation Found By the TextTiling Algorithm. $P_k$, WindowDiff, Segmentation Similarity, and Boundary Similarity are all presented between reference segmentation sets and segmentation sets chosen by TextTiling. The column labeled '% correct' is the percentage of files where SegReFree is directionally correct.

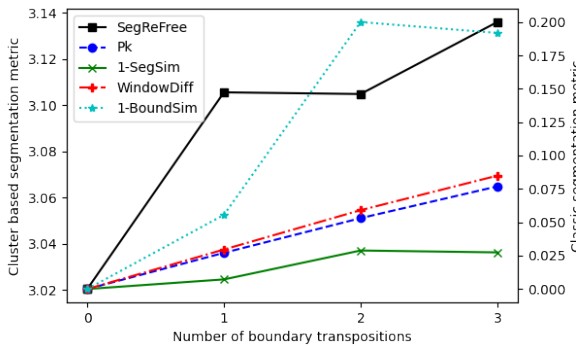

Figure 6: Proposed metric and classic metrics as a function of single random boundary transposition using the Choi 3-11 data set

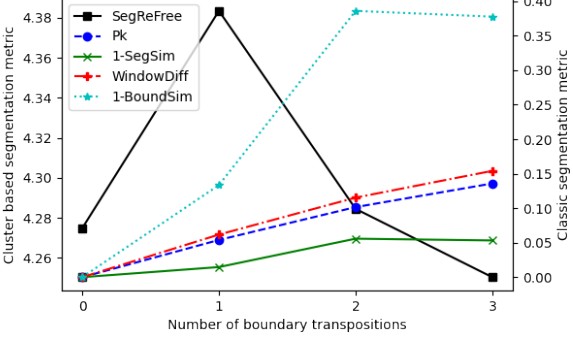

Figure 7: Proposed metric and classic metrics as a function of single random boundary transposition using the Wiki-50 data set

lected and shifted by varying numbers of sentences. As described in Section 2, this is a more challenging problem and is handled in different ways by different classic segmentation metrics. Diaz and Ouyang (2022) also notes that boundary transposition is poorly handled by other existing segmentation metrics as they treat all boundary shifts as equally bad. The mean metrics as a function of transposition distance are displayed in Figures 6 and 7 for the Choi 3-11 and Wiki-50 data sets, respectively.

A table of Pearson correlation coefficients computed between the proposed metric and classic metrics is displayed as Table 3. Our proposed metric shows strong agreement with classic metrics with the Choi 3-11 data set and directionally correct agreement for the majority of transposition distances tested on the Wiki-50 data set. We hypothesize that the degraded performance on this experiment with the Wiki-50 data set comes from the inconsistent size of segments. Because section and subsection headings are given their own segments, there are many single sentence or even single word segments within this data set. The semantic meaning of a section and an adjacent subsection heading is often quite similar and when these are combined, it would likely improve (lower) the SegReFree score.

## 4.2 Using existing segmentation methods

To provide an additional demonstration of the metrics use, an older text segmentation method, Text-Tiling (Hearst, 1997) is used to segment the Choi 3-11 and Wiki-50 data sets. This method is intentionally chosen as a lower quality, but still purposeful method of segmentation. The NLTK (Bird et al., 2009) implementation of TextTiling was used for this section. Window sizes for TextTiling were chosen arbitrarily to ensure at least two segments per file, with a value of 100 words used for the Choi 3-11 data set and a window of 20 words used for the Wiki-50 data set. Results of this test are presented in Table 4 along with classic metrics

provided for reference. To interpret these results, we assume that the quality of segmentation performed by TextTiling is lower than that of the original source segmentation, and therefore, we should expect a higher (worse) SegReFree score for the TextTiling segmentation. It can be observed that our proposed metric is directionally correct the majority of the time, showing a higher (worse) score for the lower quality segmentation 98% of the time for the Choi 3-11 data set and 68% of the time for the Wiki-50 data set.

## 5 Conclusions

In this work, a new topical text segmentation evaluation metric that requires no reference segmentation set is proposed and evaluated. To our knowledge, this is the first reference-free segmentation quality metric that has been proposed. Strong correlation to existing reference-based metrics was demonstrated for the worst case errors of missing boundaries and boundary insertion in the middle of a coherent segment. A weaker, yet present correlation is demonstrated for the harder case of boundary transposition (so-called "near misses"). The use of our reference-free metric can help enable future segmentation efforts on data that do not have existing human annotation.

## Limitations

This work was performed using a somewhat limited set of data, as the majority of text segmentation data sets found also derived from Wikipedia and were very similar to the Wiki-50 data set. It is likely that this metric may not be the best possible metric for certain segmentation use cases, such as situations where the 'ideal' segmentation has boundaries that do not correlate to semantic shifts in topic.

It is also likely that for data sets that have segments including many chunks with varying semantic meaning, such as a transcription that includes every small interjection, that this metric will lose some sensitivity and be less useful. The Wiki-50 data set is closest to this, with the inclusion of short section titles, which is why we believe we see worse performance with that data set. However, due to the lack of data sets clearly representing these specific cases, the exact impact of non-existant semantic shifts and semantically-scattered inclusions is not quantified.

It should also be noted that because this metric relies on computing distances based on semantic meaning, the SegReFree score between two different texts has no particular meaning or use. It should be used as a way to compare segmentations between consistent text or sets of texts rather than comparing segmentations on dissimilar texts.

## Ethics Statement

As this is not a generative task, the authors have no concerns about the content created by this metric. Misuse of the metric is still a possibility, such as blind reliance upon it when used for important decisions. Additionally, there may be ways for the metric to introduce bias in its usage, particularly if the training of the embedding model is not appropriate for the domain. It has been shown that contextual word representations, such as BERT, can carry biases (Kurita et al., 2019); so it is also likely that bias present in Sentence-BERT would propagate into models dependent on it, such as this one.

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

# A   Appendix A: Human segmentation comparison with Stargazer Data Set

To provide an additional demonstration of the utility of the proposed metric, the Stargazer segmentation data set is evaluated. The Stargazer article (Baker, 1990) was selected as a challenging segmentation task for human annotators and (Hearst, 1997) collected seven different human segmentation's with the intent of developing segmentation agreement metrics. Our proposed metric can be used to evaluate the topical segmentation of the different individual metrics. To provide a reference, the best scoring annotation found with SegReFree

| Annotator | SegReFree Score | $P_k$ relative to annot. 4 |
|-----------|-----------------|----------------------------|
| 1 | 3.15 | 0.42 |
| 2 | 3.82 | 0.47 |
| 3 | 2.32 | 0.26 |
| 4 | 2.23 | 0 |
| 5 | 3.66 | 0.37 |
| 6 | 3.67 | 0.47 |
| 7 | 3.52 | 0.32 |

Table 5: Proposed metric computed for Stargazer segmentation data set

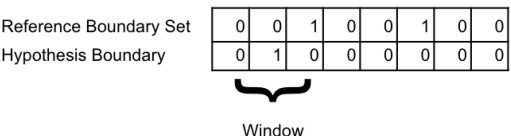

Figure 8: Window-based segmentation comparison

Figure 9: Mass-based segmentation comparison

is chosen as a hypothesis and compared with each of the other annotations using the $P_k$ metric. The two metrics correlate fairly well, with the highest $P_k$ score aligned with the two largest SegReFree scores found.

## B  Appendix B: Justification for data set selection

Our choice to only use two segmentation data sets is based on similar numbers of real language data sets (or fewer) used in the literature that we cite in this work. Fournier and Inkpen (2012) 2012 used one natural language data set for one subsection, Pevzner and Hearst (2002) did not include any natural language data sets; Both Solbiati et al. (2021) and Ghinassi (2021) used two data sets for evaluation of their segmentation methods. Instead of a wide diversity of data sets as may be found in other research topics, we focus on presenting a diverse set of experiments in Section 4.

## C  Appendix C: Visualization of window and mass based segmentation

A simple visualization of a window-based segmentation comparison is shown in Figure 8. Here, the start of a new segment is represented by a *1*. The window-based metrics use a sliding window that compares segment boundaries between reference and hypothesis sets within the window. The same example segmentation set is presented in mass-based format in Figure 9, where the segments are represented as alternating shaded boxes. In a mass-based metric, the sizes of the segments in terms of chunks are used as the basis for comparison.

## D  Appendix D: Ablation study

To validate the design choices used in our proposed metric, a small ablation study was performed using both data sets and both of the synthetic degraded segmentation experiments. As with previous experiments, random selection of boundaries was performed several times (five for each file in the Choi 3-11 data set and 30 for each file in the Wiki-50 data set) and results were averaged. The two modifications made to the original Davies-Bouldin Index are described in detail in Section 3.3 and are the inclusion of a size penalty term as well as an adjacency requirement for the computation of the metric. Table 6 shows the results of the removal of these individual alterations by computing the Pearson correlation coefficient between the Davies-Bouldin based metric and the $P_k$ metric. The $P_k$ metric was chosen as a comparison due to it's position as the oldest and most common segmentation metric. It can be seen that the original Davies-Bouldin Index does not correlate with $P_k$ as well as our proposed metric does, and for the some experiments has a negative correlation. The size penalty appears to be helpful for improving performance in the presence of segment boundary additions (simulated as segment splits), which causes smaller-than-desired segments. The adjacency appears to be most necessary for penalizing missed (simulated as deleted) segment boundaries, which conceptually makes sense as a missed boundary would cause two different topical segments to become combined.

Multiple distance metrics were also tested, including L1, L2, and Cosine distance. Although L1 distance slightly outperformed L2 distance in a significant way on one experiment, L2 distance was more used in Davies-Bouldin Index literature and was chosen for our proposed metric.

| Ablation | Choi Split | Choi Del. | Wiki-50 Split | Wiki-50 Del. |
|---|---|---|---|---|
| Proposed | 0.999 | 0.998 | 0.694 | 0.985 |
| -Size Penalty | 0.988 | 0.999 | -0.994 | 0.994 |
| -Adjacen. Req. | 0.816 | -0.844 | 0.940 | 0.960 |
| Orig. DBI | 0.763 | -0.844 | -0.998 | 0.993 |
| L1 Dist. | 0.999 | 0.998 | 0.732 | 0.986 |
| Cos. Dist. | -0.871 | -0.832 | -0.806 | 0.987 |

Table 6: Correlation Between Ablated Versions of Proposed Metric and $P_K$.
Original DBI includes neither adjacency requirement nor size penalty term. Proposed metric uses L2 distance.