# OpenReview forum: "A Reference-free Segmentation Quality Index (SegReFree)"
_EMNLP/2023/Conference — EMNLP 2023 Findings_

### Official Review · Reviewer_8kNW · 2023-07-24

**Soundness:** 4

**Excitement:**

4: Strong: This paper deepens the understanding of some phenomenon or lowers the barriers to an existing research direction.

**Missing References:**

Gerardo Ocampo Diaz and Jessica Ouyang. An alignment-based approach to text segmentation similarity scoring. CONLL 2022.

**Paper Topic And Main Contributions:**

The paper presents a reference-free topic segmentation metric. Existing metrics require a reference and so cannot be used on data without gold segmentations. The proposed "SegReFree" metric is based on the Davies-Bouldin Index for cluster validity.

An embedding is computed for each sentence in the document using Sentence-BERT, and all embeddings are labeled with the segment (cluster) their sentence belongs to in the candidate segmentation. The authors make two modifications to the Davies-Bouldin Index to make it more suitable for sequence segmentation. First, they only compare segments that are adjacent in the document, since topic segmentation is only concerned with placing boundaries between adjacent segments. Second, the average intra-segment similarity score is penalized exponentially based on the number of sentences in the segment to prevent small segments with very few sentences from receiving significantly better scores than larger segments; the authors note that adjacent sentences can be expected to be more similar than sentences farther apart, even within a segment.

The authors perform experiments on two topic segmentation datasets, demonstrating strong correlations between SegReFree and existing metrics for boundary removal and segment splitting errors. The results on boundary transposition errors are more mixed. Prior work has noted that transpositions can be treated very differently among the existing metrics (see missing references), so these results seem reasonable.

**Reasons To Accept:**

The proposed metric shows strong correlation with existing segmentation metrics for most error types and can be used without a reference segmentation, which existing metrics require. The motivation behind the metric is clearly explained, its design is sensible, and it will be useful to the community, especially for researchers wanting to evaluate segmentation as an upstream task, who are unlikely to have reference segmentations available.

**Reasons To Reject:**

The modifications made to the Davies-Bouldin Index are relatively minor and straightforward.

**Reproducibility:**

4: Could mostly reproduce the results, but there may be some variation because of sample variance or minor variations in their interpretation of the protocol or method.

**Reviewer Confidence:**

4: Quite sure. I tried to check the important points carefully. It's unlikely, though conceivable, that I missed something that should affect my ratings.

**Typos Grammar Style And Presentation Improvements:**

Section 2.1, Line 103: The visualization mentioned in this line is missing from the Appendix (reference link is also broken).
Section 2.1.1, Line 117: Figure is missing, and reference link is broken.
Section 2.1.2, Line 138: Same as above.

---

### Official Review · Reviewer_Cxez · 2023-07-28

**Soundness:** 3

**Excitement:**

3: Ambivalent: It has merits (e.g., it reports state-of-the-art results, the idea is nice), but there are key weaknesses (e.g., it describes incremental work), and it can significantly benefit from another round of revision. However, I won't object to accepting it if my co-reviewers champion it.

**Paper Topic And Main Contributions:**

This paper describes a metric to assess the segmentation quality of a text. As input, the metric reads a topic-segmented text and as output, it returns a quality index. The lower the index, the better the quality. The index is unsupervised and uses the SBERT transformer to vectorize the sentences in a segment. The rationale is that that sentences in a segment should be close to each other and two sentences in two different segments should be away from each other. For each segment, the algorithm computes the centroid of the SBERT embeddings of the sentences in this segment. Then the algorithm computes a ratio of pairwise intracluster segments for adjacent segments. This method is a variant of the Davies-Boulding Index.
The authors assess their metric by degrading already segmented texts. The degradation was obtained by either removing boundaries or splitting segments. On the two evaluation datasets, the proposed metric revealed a consistent behavior reflecting the degradation and proved better than baseline techniques.

**Questions For The Authors:**

1/ How Solbiati et al. (2021) and Ghinassi (2021) is different from what you proposed
2/ Why don't you use your model to segment a text from scratch?

**Reasons To Accept:**

1/ A simple model that measures a text segmentation quality
2/ Convincing experiments showing that the index can reflect well the degradation of a segmentation
2/ An evaluation on two datasets and a comparison with relatively old baseline algorithms where the index proved better that other baseline techniques

**Reasons To Reject:**

1/ The model is not compared with newer techniques using embeddings
2/ The authors do not write how Solbiati et al. (2021) and Ghinassi (2021) is different from what they proposed
3/ A few issues in the formatting of the submitted version: missing figures and crossreferences

**Reproducibility:**

5: Could easily reproduce the results.

**Reviewer Confidence:**

4: Quite sure. I tried to check the important points carefully. It's unlikely, though conceivable, that I missed something that should affect my ratings.

**Typos Grammar Style And Presentation Improvements:**

Figure 1 is missing from the paper and some cross-references are replaced by ??.

---

> ### Author Rebuttal · Authors · 2023-08-29
>
> Thank you for your detailed review.
>
> We will fix the cross-references for the camera ready version, thank you for pointing them out.
>
> Regarding the Questions For The Authors:
>
> 1. "How Solbiati et al. (2021) and Ghinassi (2021) is different from what you proposed"
>
> This is explained on lines 243-257 - they use sentence embeddings to create a segmentation algorithm, we propose a segmentation metric agnostic of method. Our proposed method could be used for comparing human annotators, for example, which is not doable with a segmentation algorithm.
>
> 2. Why don't you use your model to segment a text from scratch?
>
> Our goal is to create a segmentation metric, not a segmentation algorithm. We could utilize it with a greedy algorithm to segment a text, but that is not the design goal.
>
> Regarding the Reasons To Reject:
>
> 1. The model is not compared with newer techniques using embeddings
>
> We assume you are referring to Solbiati (2021) and Ghinassi (2021). Neither is a segmentation metric (both are segmentation algorithms) and therefore not comparable, like WindowDiff/Pk/etc..
>
> 2. The authors do not write how Solbiati et al. (2021) and Ghinassi (2021) is different from what they proposed
>
> This is explained on lines 243-257 - they use sentence embeddings to create a segmentation algorithm, we propose a segmentation metric agnostic of method. Neither is a segmentation metric (both are segmentation algorithms) and therefore not comparable, like WindowDiff/Pk/etc..
>
> 3. A few issues in the formatting of the submitted version: missing figures and crossreferences
>
> Thank you for pointing these out - they will be fixed in the camera ready version.

---

### Official Review · Reviewer_ZTcw · 2023-08-05

**Typos Grammar Style And Presentation Improvements:** 1. References to figure and appendice…
**Soundness:** 4

**Excitement:**

3: Ambivalent: It has merits (e.g., it reports state-of-the-art results, the idea is nice), but there are key weaknesses (e.g., it describes incremental work), and it can significantly benefit from another round of revision. However, I won't object to accepting it if my co-reviewers champion it.

**Paper Topic And Main Contributions:**

This work proposes a new reference-free segmentation metric that does not need to rely on reference or human-labeled segmentation to evaluate the quality by using sentence embedding and modifying a cluster validity metric. On two datasets, the authors show high correlations with classical reference-based segmentation metrics in a variety of settings.

**Questions For The Authors:**

1. Does the metric contain any bias for segmentation using SentenceTransformers, since it is used as part of the metric?
2. Have you explored incorporating this metric to improve segmentation quality?

**Reasons To Accept:**

1. The paper is clearly written and provides a comprehensive introduction and related work on the topic of segmentation.
2. The authors provide strong results indicating the effectiveness of their evaluation metric which highly correlates with reference-based indices.

**Reasons To Reject:**

1. It is sometimes hard to understand what to make of the experiments. For example, what is 1-seg-sim and 1-bound-sim? What is the takeaway of all the figures? Should any one metric achieve low scores throughout?
2. The paper would benefit more from polishing the presentation of the paper.

**Reproducibility:**

5: Could easily reproduce the results.

**Reviewer Confidence:**

3: Pretty sure, but there's a chance I missed something. Although I have a good feel for this area in general, I did not carefully check the paper's details, e.g., the math, experimental design, or novelty.

---

> ### Author Rebuttal · Authors · 2023-08-29
>
> Thank you for your detailed review.
>
> We apologize for the missing reference links to figures, that will be fixed in the camera ready version. We will also update Figure 1, add periods after captions, and try combining some tables and figures.
>
> Regarding the reasons to reject:
>
> 1.1 "It is sometimes hard to understand what to make of the experiments."
>
> The goal of the experiments is to demonstrate the strong correlation between our proposed metric and the existing metrics as they respond to realistic segmentation errors. This is articulated in Section 4.1 and 4.2. We can also add language to make this more explicit when referencing Figures 2-7.
>
> 1.2 "For example, what is 1-seg-sim and 1-bound-sim?"
>
> In Section 2.1, where existing metrics are introduced, we discuss how Pk and WindowDiff have a zero as an optimal score and Segmentation Similarity and Boundary Similarity have optimal scores of 1; because of this we plot 1 minus the Segmentation Similarity and Boundary Similarity to make the trends consistent. This is explicitly stated in the sub-caption of Tables 1-3, but we can add a sentence to the camera ready version to make this more clear when referencing the figures.
>
> 1.3 "What is the takeaway of all the figures? Should any one metric achieve low scores throughout?"
>
> The takeaway of the figures is that we show that our proposed segmentation metric (SegReFree) responds to segmentation errors in a way that is similar to existing segmentation metrics. Ideally, all segmentation metrics should perform similarly.
>
> 2. The paper would benefit more from polishing the presentation of the paper.
>
> Thank you for your input. We feel that after updating the missing reference links, adding clarifying language, and adding the reference suggested by Reviewer 3, we will have a nicely polished paper.
>
>
> Regarding your Questions For The Authors:
>
> 1. Does the metric contain any bias for segmentation using SentenceTransformers, since it is used as part of the metric?
>
> It is likely that there are very abstract influences of bias on the segmentation performance because of the use of SentenceTransformers. It is known that contextual word embeddings show different social biases [1], so it is likely that a sentence level embedding would learn bias from its training data. How this bias would affect segmentation is a harder question - figuring this out is a question probably spanning multiple papers. Thank you for asking this question, we will add some of this discussion to our Ethics Statement.
>
>
> 2. Have you explored incorporating this metric to improve segmentation quality?
>
> In internal experiments, we have used this metric to tune segmentation algorithms for use in a summarization task. Due to the use of the internal dataset, we have not published these experiments. We feel that the use of this algorithm for summarization would fit into another paper, as optimal segmentation for summarization is an area of open research, such as [2].
>
> [1] Kurita, Keita, et al. "Quantifying social biases in contextual word representations." 1st ACL Workshop on Gender Bias for Natural Language Processing. 2019.
>
> [2] Zhang, Yusen, et al. "SummN: A Multi-Stage Summarization Framework for Long Input Dialogues and Documents: A Multi-Stage Summarization Framework for Long Input Dialogues and Documents." Proceedings of the 60th Annual Meeting of the Association for Computational Linguistics (Volume 1: Long Papers). 2022.

---

### Meta-Review · Area_Chair_NsNv · 2023-09-15

**Recommendation:** 3

**Metareview:**

This paper proposes a new metric for summarization, which does not require manual annotations. The reviewers agree about the soundness of the submission but are split between "strong" and "ambivalent" on excitement. There is some debate about the relationship between reference-free segmentation *evaluation* and segmentation *algorithms*, since the former seems to imply the latter. Another excitement issue is the degree of innovation over the Davies-Bouldin segmentation algorithm, which is the foundation for the approach.

---

### Decision · Program_Chairs · 2023-10-07

**Decision:**

Accept-Findings

**Comment:**

This paper proposes a new metric for summarization, which does not require manual annotations. The reviewers agree about the soundness of the submission but are split between "strong" and "ambivalent" on excitement. There is some debate about the relationship between reference-free segmentation *evaluation* and segmentation *algorithms*, since the former seems to imply the latter. Another excitement issue is the degree of innovation over the Davies-Bouldin segmentation algorithm, which is the foundation for the approach.